# Two-dimensional electronic spectroscopy reveals liquid-like lineshape dynamics in CsPbI$_3$ perovskite nanocrystals

Hélène Seiler [1,2], Samuel Palato[1,2], Colin Sonnichsen[1], Harry Baker[1], Etienne Socie[1,3], Dallas P. Strandell[1] & Patanjali Kambhampati[1]*

Lead-halide perovskites have attracted tremendous attention, initially for their performance in thin film photovoltaics, and more recently for a variety of remarkable optical properties. Defect tolerance through polaron formation within the ionic lattice is a key aspect of these materials. Polaron formation arises from the dynamical coupling of atomic fluctuations to electronic states. Measuring the properties of these fluctuations is therefore essential in light of potential optoelectronic applications. Here we apply two-dimensional electronic spectroscopy (2DES) to probe the timescale and amplitude of the electronic gap correlations in CsPbI$_3$ perovskite nanocrystals via homogeneous lineshape dynamics. The 2DES data reveal irreversible, diffusive dynamics that are qualitatively inconsistent with the coherent dynamics in covalent solids such as CdSe quantum dots. In contrast, these dynamics are consistent with liquid-like structural dynamics on the 100 femtosecond timescale. These dynamics are assigned to the optical signature of polaron formation, the conceptual solid-state analogue of solvation.

[1] Department of Chemistry, McGill University, 801 Sherbrooke St. West, Montréal, Québec, Canada. [2] Present address: Department of Physical Chemistry, Fritz-Haber-Institut der Max-Planck-Gesellschaft, Faradayweg 4-6, 14195 Berlin, Germany. [3] Present address: Lausanne Centre for Ultrafast Science (LACUS), École polytechnique fédérale de Lausanne, 1015 Lausanne, Switzerland. *email: pat.kambhampati@mcgill.ca

Lead-halide perovskites of the type APbX₃, where A is an inorganic or organic cation and X is a halide, have recently attracted significant attention for their potential in optoelectronic applications, in particular photovoltaics[1–4]. In addition to a large absorption cross section in the visible range, high carrier mobility, and low non-radiative recombination rates, lead-halide perovskites also possess structural properties that qualitatively distinguish them from other inorganic semiconductors. It is the interplay of these electronic and structural properties that give rise to the performance of lead-halide perovskites[5]. How the structural properties impact the optoelectronic performance, and the role of dynamic disorder in protecting the optical excitation from scattering processes, is a topic of intense investigation.

In order to understand the relationship between structure and function in lead-halide perovskites, a variety of methods have recently been applied. Collectively, these methods reveal a picture consistent with structural dynamics following photoexcitation[6–12]. The dynamics have been assigned to polaron formation and described in terms of a phonon glass with a localized ultrafast response to optical excitation[6,7]. Polaron formation arises from the dynamical coupling of atomic structure fluctuations to electronic states[13,14]. This results in a fluctuating electronic gap. Understanding the properties of these fluctuations is essential since it dictates the electronic response of interest for optoelectronic applications. Previous dynamical probes have focused on the terahertz and Raman responses, and have shown how specific phonons are involved in the polaron formation process[6,7,10–12]. For the time-resolved electronic response, one can measure transient-absorption (TA) spectra in the visible region[15]. With two-dimensional electronic spectroscopy, one can additionally measure lineshape dynamics.

Here we employ 2DES to measure the homogeneous lineshape dynamics of CsPbI₃ nanocrystals. These dynamics reveal an isomorphism between the lineshape dynamics of a molecular dye undergoing solvation dynamics and that of CsPbI₃ nanocrystals. We assign the observed dynamics in the perovskites to polaron formation, as a conceptual analogue of solvation. By virtue of spreading the optical response on a 2D correlation map, 2DES clearly distinguishes sources of static and dynamic disorder with ~10 fs time resolution. These lineshape dynamics are unique to the 2DES method and cannot be accessed by one-dimensional methods, such as TA spectroscopies. Our measurement yields polaron formation time and binding energy, providing a complement to previous experimental studies and a direct point of comparison for future ab initio works. At low fluences, nanocrystals enable working in the single excitation per particle regime, thereby minimizing effects arising from electron–electron or hole–hole scattering processes[16]. The use of nanocrystalline perovskites also enables comparison with model systems such as covalent polar nanocrystal quantum dots and molecular dyes in solution. Our data show that the homogeneous linewidth of the CsPbI₃ nanocrystals evolves on the 100-fs timescale in a manner consistent with ultrafast polar solvation dynamics, contrasting with the response of covalent CdSe nanocrystal quantum dots. We demonstrate excellent agreement of the experimental perovskite data with modeled data for dissipative, collective reorganization of the lattice, which enables us to assign the observed spectral dynamics to polaron formation.

## Results

**Static measurements.** CsPbI₃ nanocrystals were synthesized following previously described methods[2,17]. Details are provided in the Methods section. Figure 1 displays an overview of the basic properties of these CsPbI₃ nanocrystals. The linear absorption spectrum is shown in Fig. 1a, together with the photoluminescence spectrum. The nanocrystals show weakly confined excitons, by virtue of their large size relative to the Bohr radius. For smaller sizes, one finds an excitonic spectrum consistent with the strongly confined quantum dot regime[2,18]. A typical transmission electron microscope image is shown in the inset, from which we estimate that the cubic-shaped CsPbI₃ nanocrystals have an average size of 9.8 nm (Supplementary Fig. 2). A static X-ray diffraction pattern is shown in Fig. 1b. The Bragg peaks are broad due to the small size of the nanocrystals[19]. These simple properties are consistent with a variety of lead-halide perosvkite nanocrystals[1,20,21]. Figure 1c, d displays cartoons of phonons (c) and a polaron (d), both of which are expected to play important roles in electron relaxation following photoexcitation.

**2DES measurements.** In order to directly probe the electronic structure and dynamics of the CsPbI₃ nanocrystals, we employ the 2DES pulse sequence shown in Fig. 2a. We refer the reader to references for a more detailed overview of the technique[22,23]. Implementation details are provided in the Methods section and elsewhere[24]. Figure 2b, c displays two representative correlation maps obtained on the CsPbI₃ perovskite nanocrystals, at $t_2 = 20$ and 600 fs. The main qualitative observation is the transformation of the 2D lineshape from elliptical to circular. This behavior is representative of spectral diffusion, a classic example being solvation dynamics[25–27]. The timescale of lineshape dynamics informs on how fast the environment responds to the charge rearrangement resulting from the optical excitation. Figure 2d, e illustrates the basic concepts of spectral diffusion. Figure 2d shows the time-dependent frequency gap, $\omega + \delta\omega(t_2)$, where $\delta\omega(t_2)$ represents the equilibrium fluctuations from the mean transition frequency. The charge rearrangement due to a single optical excitation is a perturbation to the equilibrium charge distribution. Hence, by the fluctuation–dissipation theorem, the nonequilibrium response of the environment is given by the correlation function of the equilibrium fluctuations[28]. The equilibration of the electronically excited state is measured via the frequency–frequency correlation function (FFCF), $\langle\delta\omega(t_2)\delta\omega(0)\rangle$, which decays with the environment's response time constant $\tau_c$, as shown in Fig. 2e. The FFCF relates to the experimental observable of the time-dependent energy gap in 1D spectroscopy. In 2D spectroscopy, one is able to observe diagonal and anti-diagonal linewidths that reflect the process of spectral diffusion with less ambiguity.

**Comparative analysis.** In order to place the spectral dynamics of the CsPbI₃ perovskites in terms of model systems at two qualitative limits, we conduct a comparative analysis on CdSe nanocrystals and a molecular dye in solution. The key results are summarized in Fig. 3. Figure 3a–c displays a series of spectral projections as a function of $t_2$, obtained from the 2D spectra by selecting a value of $E_1$ (pump energy) and reporting the spectral projection as a function of $E_3$ (probe energy). These datasets can be interpreted in a manner similar to a TA spectrum. Figure 3a shows that the early-time dynamics in the CsPbI₃ nanocrystals are dominated by continuous energy relaxation. These data are in stark contrast with the behavior of the CdSe nanocrystals, shown in Fig. 3b, where the spectral dynamics are well known to arise from relaxation within a discrete manifold of states[29]. Furthermore, the observed dynamics for CsPbI₃ are slow compared with the molecule in solution, shown in Fig. 3c, where a similar continuous energy relaxation is expected from solvation but is not captured by our temporal resolution and spectral bandwidth. Despite the high temporal and energy resolution of these spectral projections, their interpretation is complex due to the presence of

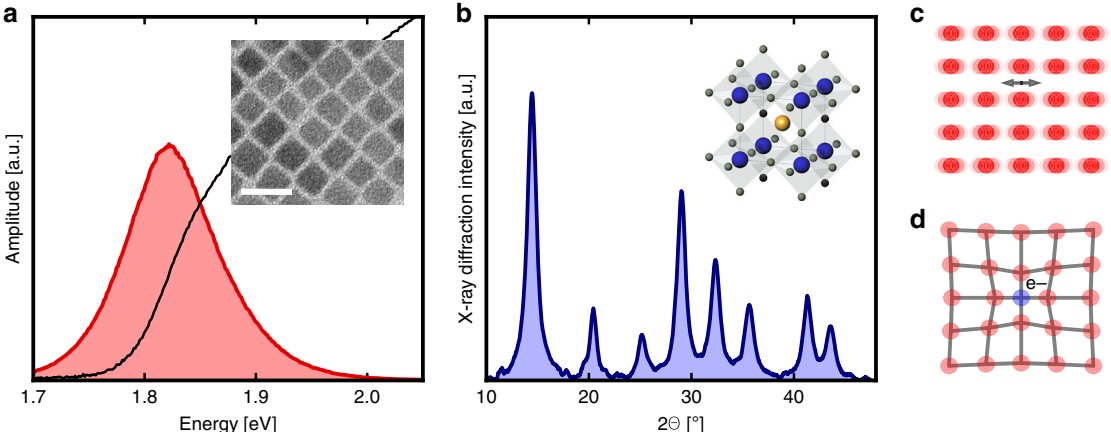

**Fig. 1** Static measurements of optical and structural properties of CsPbI$_3$ nanocrystals. **a** Linear optical absorption (black) and photoluminescence (red) of the perovskite nanocrystals in solution. Inset: representative transmission electron microscope image of the sample under study. Scale bar: 20 nm. **b** X-ray diffraction data of the nanocrystals (dark blue). **c**, **d** Cartoon representations of phonons (**c**) and a polaron (**d**)

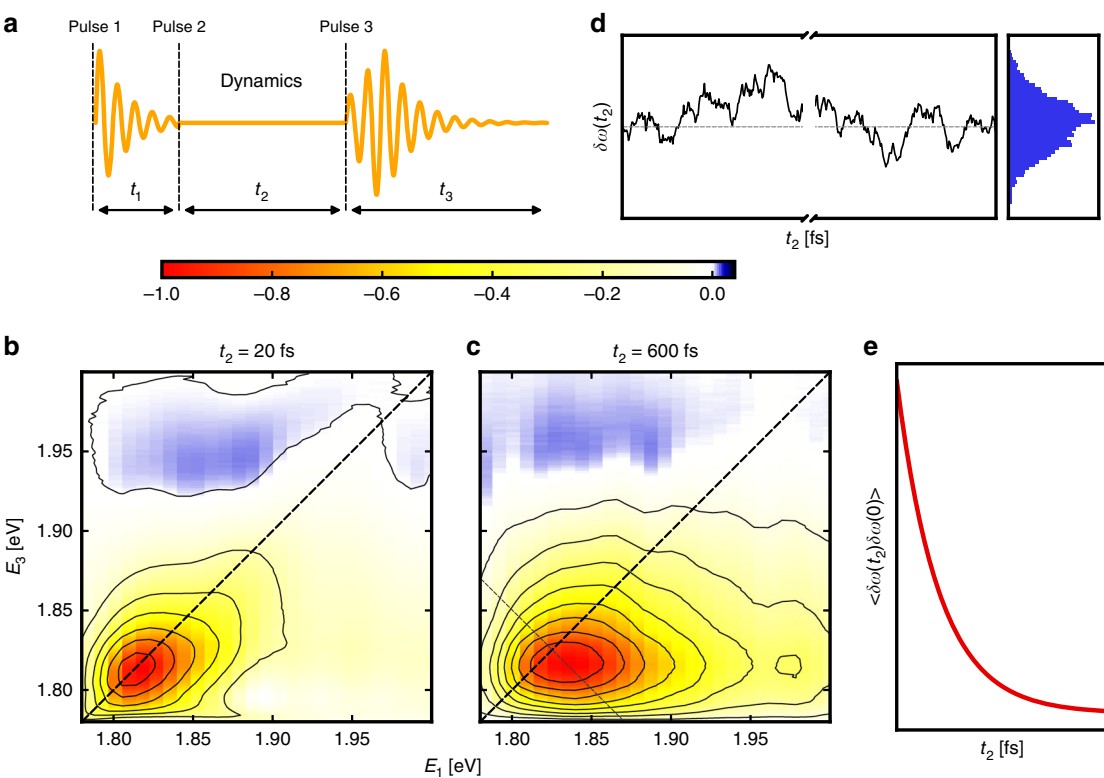

**Fig. 2** 2DES measurements on the CsPbI$_3$ nanocrystals and basic concepts of spectral diffusion. **a** Typical pulse sequence in a 2D experiment. Pulse 1 creates a coherent superposition of quantum states, which dephases during $t_1$. Pulse 2 transforms the coherence into a population state, which undergoes dynamics. Pulse 3 creates a coherence again, and the emitted field is detected. **b**, **c** Exemplary 2D spectra of an ensemble of CsPbI$_3$ nanocrystals, for $t_2 =$ 20 fs (**b**) and $t_2 = 600$ fs (**c**). **d** The rounding up of the peak in the 2D spectrum as $t_2$ increases can be described by the process of spectral diffusion. Here a simulated Ornstein–Uhlenbeck trajectory illustrates the modulation of the bandgap by correlated fluctuations. The histogram shows the distribution of frequencies, which approaches a Gaussian upon memory loss of the initial frequency. **e** The frequency–frequency correlation function describes the loss of correlation as a function of $t_2$

many overlapping contributions. Indeed, the fast red shift followed by a blue shift observed in Fig. 3a is indicative of competition between mechanisms, such as screening and population cooling.

A closer inspection of the spectral projections shown in Fig. 3a is obtained by reporting pseudo-TA spectra for a set of delays,

shown in Fig. 4. These spectra can be analyzed in terms of ground-state bleach, stimulated emission, and excited-state absorption contributions, just like in a TA experiment. The only difference is the additional presence of electronic coherences, which are neglected here. The spectra in Fig. 4a show the rise of a tail to the blue of the main peak, indicated by the black arrow.

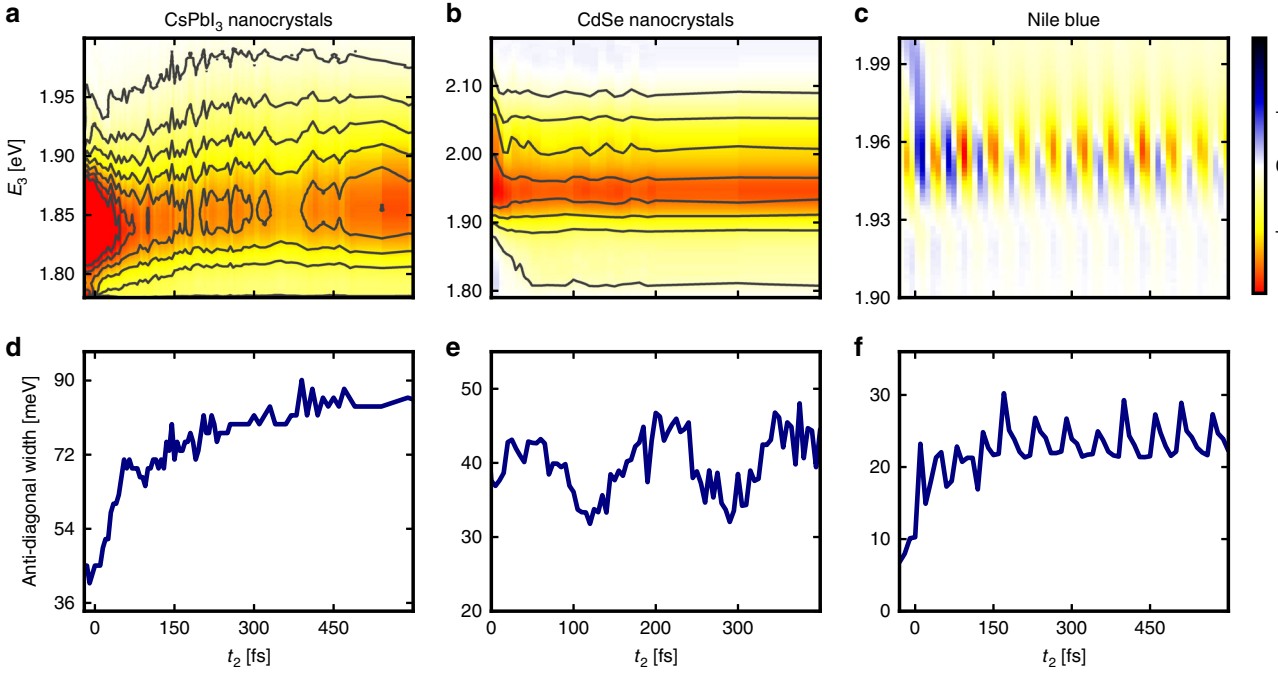

**Fig. 3** Comparative analysis shows liquid-like behavior of the perovskite nanocrystals. **a–c** Pseudo-TA projections obtained by selecting a value of $E_1$ (narrow pump) on the 2D spectrum and reporting the spectral projections as a function of $E_3$ (broadband probe). The CsPbI$_3$ nanocrystals' pseudo-TA map (**a**) is shown for $E_1 = 1.82$ eV. The CdSe nanocrystals' (**b**) and Nile Blue (**c**) pseudo-TA maps are both shown for $E_1 = 1.95$ eV. **d–f** Extracted anti-diagonal linewidth from the 2D spectra shown as a function of $t_2$ for CsPbI$_3$ (**d**), CdSe (**e**), and Nile Blue (**f**)

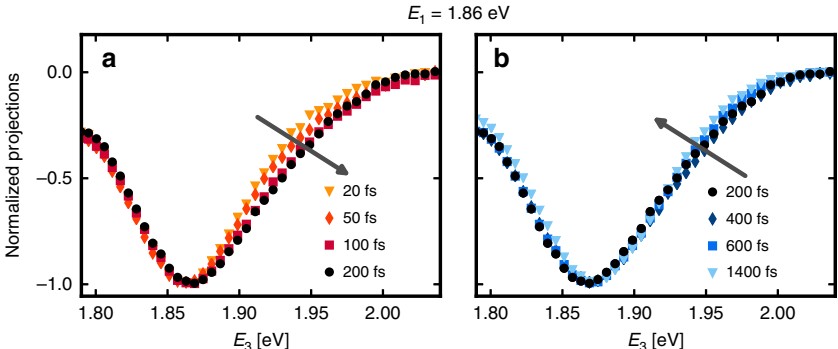

**Fig. 4** Pseudo-TA spectra can be extracted from the 2D dataset. **a** The early-time projections show the rise of a tail to the blue as $t_2$ increases, inconsistent with previously reported cooling signatures. **b** The tail is observed to slightly decrease at later population times, indicative of the onset of cooling

This tail slightly relaxes at later population times, as can be seen in Fig. 4b. The early-time dynamics of the spectra are inconsistent with previous cooling signatures and confirm that other phenomena dominate the spectral changes[15,30]. No clear qualitative picture emerges from these one-dimensional results.

In order to unravel the spectral diffusion dynamics in greater detail, we focus on the unique observation enabled by the 2DES method: the time dependence of the anti-diagonal linewidth, represented as the dashed gray line in Fig. 2c. Details on how to extract the anti-diagonal linewidth are presented in Supplementary Note 2 as well as Supplementary Figures 8 and 9. We note that the anti-diagonal linewidth is only a true representation of the homogeneous linewidth in the limit of strong inhomogeneous broadening. Due to the size dispersion of the nanocrystals (see Supplementary Fig. 2), we can expect a large static disorder contribution to the diagonal linewidth of the 2D peak. Here we report the full-width at half-maximum (FWHM) of the anti-diagonal lineshape as a function of $t_2$, which has the advantage to be model free but is an approximation to the true homogeneous linewidth. Figure 3d shows that the anti-diagonal width of the CsPbI$_3$ nanocrystals quickly broadens from its initial homogeneous value to its final value at 400 fs. In contrast, as seen in Fig. 3e, the width of the CdSe nanocrystals stays constant on average, indicating that no spectral diffusion is taking place. Instead, the width is modulated at the LO phonon frequency. Figure 3f shows the lineshape dynamics for Nile Blue in ethanol. By comparing Fig. 3d, f, it becomes apparent that the lineshape behavior of the CsPbI$_3$ nanocrystals qualitatively mimics that of the molecular dye in solution, albeit with a slower timescale and without the strong coherent vibronic modulations arising from the ring distortion mode of Nile Blue[31]. These data reveal that perovskite nanocrystals undergo spectral dynamics that are consistent with liquids and inconsistent with covalent solids.

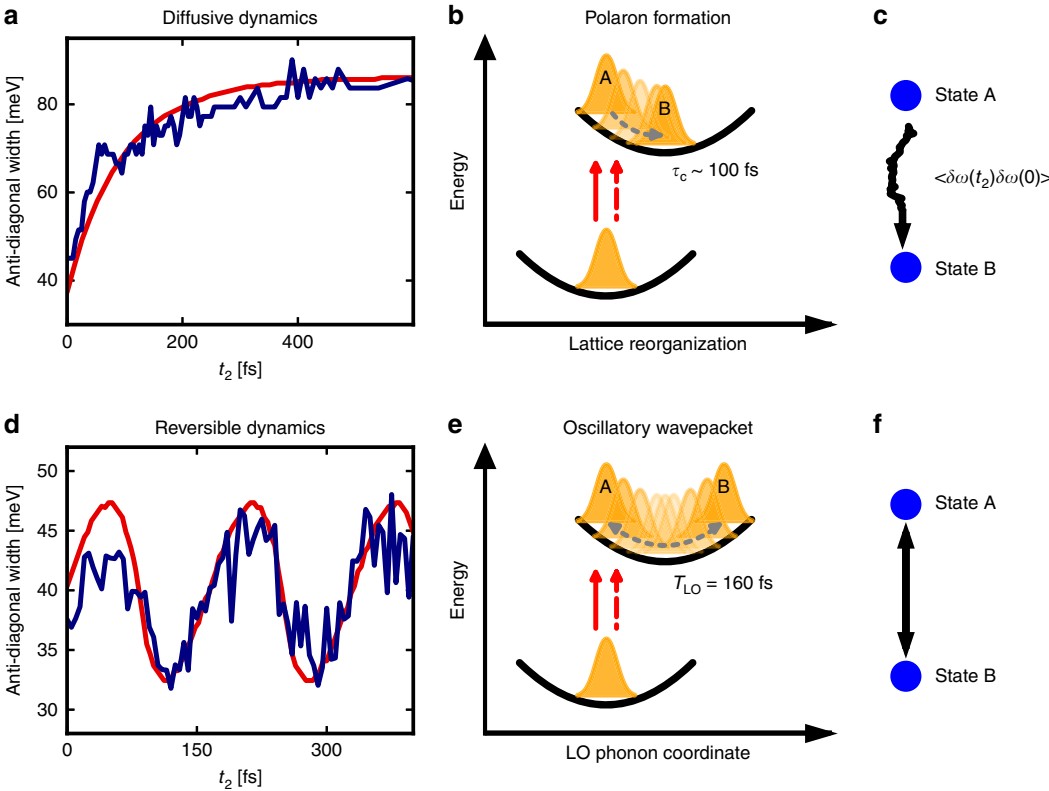

**Fig. 5** Modeling of the CsPbI$_3$ nanocrystals' lineshape evolution by diffusive dynamics. **a** The anti-diagonal linewidth of the CsPbI$_3$ nanocrystals (dark-blue curve) is well modeled by a Kubo lineshape (red curve), corresponding to dissipative dynamics typical of molecular solvation or solid-state polaron formation. **b, c** Schematic illustrations of the dynamics in (**a**). Collective structural dynamics following photoexcitation (red arrows), whether solvent or lattice, result in excited-state dynamics (orange wavepackets) that evolve from an initial state A toward equilibrium B. **d** The anti-diagonal linewidth of the CdSe nanocrystals (dark-blue curve) is well modeled by the Huang–Rhys regime (red curve), corresponding to electron–phonon coupling with quantum modes undergoing reversible wavepacket dynamics with period $T_{LO}$. **e, f** Schematic illustrations of the dynamics in (**d**), with a single quantum coordinate. This regime supports wavepacket dynamics, where the electron wavepacket (in orange) oscillates between states A and B

## Discussion

Figure 5 rationalizes the observed linewidth behaviors in light of two limits. The first limit corresponds to the diffusive regime, described by irreversible dynamics characteristic of liquids, glasses, and other disordered systems[7,25,26]. Figure 5a–c shows the dynamics of the CsPbI$_3$ perovskites along with diffusive modeling. The second limit corresponds to reversible wavepacket dynamics, characteristic of vibronic quantum systems. Figure 5d–f shows the dynamics of CdSe nanocrystals along with coherent modeling. In order to reproduce the lineshape behaviors, we performed calculations of the 2D spectra by using the cumulant expansion to second order and the multimode Brownian oscillator model[22,23,32]. Details can be found in the Supplementary Discussion. The 2D spectra of the CsPbI$_3$ and CdSe nanocrystals were modeled by using a Kubo and a Huang–Rhys lineshape, respectively[33]. Parameter values are summarized in Supplementary Tables 2 and 3. The CsPbI$_3$ dynamics are well captured by a dissipative, diffusive model, illustrated schematically in Fig. 5b, c.

The Kubo lineshape is derived from the fluctuation–dissipation theorem, with the assumption of Gaussian fluctuations with finite correlation time. It is well known that solvation in a polar solvent can be cast as a fluctuation–dissipation problem[25–27]. The conceptual analogy between solvation in liquids and polaron formation in solids has been established in other studies, such as photoemission works and more recent optical Kerr effect measurements[6,34]. Indeed, both solvation and polaron formation arise from atomic fluctuations[13,14]. While the microscopic nature of these fluctuations are not obviously connected with each other in the two cases, both are expected to yield isomorphic diffusive dynamics. For polar solvation dynamics, one has an electronic system coupled to a diffusive solvent bath. For polarons in these bulk-like nanocrystals, this system-bath partitioning is not as straightforward. From liquids, to glasses, to these ionic crystals, the dynamics reflect dynamical disorder in the system. Because properties of the fluctuations are accessed via the FFCF, the 2DES method is uniquely positioned to investigate solvation in polar liquids and polaron formation[35,36]. Combining the knowledge from previous studies probing structural dynamics in lead-halide perovskites with our main observation of solvation-like electronic response of the CsPbI$_3$ nanocrystals, we assign these diffuse dynamics to the optical signature of polaron formation.

The lineshape dynamics of the full 2D dataset is well represented by a simple model with two parameters: a correlation time of the fluctuations $\tau_c$ ~110 fs, and the amplitude of the fluctuations, $\Delta$ ~50 meV. We stress that these parameters are not accessible via one-dimensional methods such as TA spectroscopy. From $\Delta$, we can furthermore retrieve a reorganization energy $\lambda = \Delta^2/2k_B T$ ~50 meV[37]. This value is a measure of polaron stability, which is expected to play an important role in the competition between polaron formation and hot electron cooling[38]. Our extracted value is on the same order of magnitude as previous cryo-magneto optical absorption spectroscopy measurements, and also falls within the bounds of calculated binding energies for hybrid organic–inorganic perovskites[7,39]. As can be seen from Supplementary Fig. 10, our fits are very sensitive to the

value of $\Delta$. Having access to the polaron binding energy can help understand why certain lead-halide perovskites feature slower or faster cooling rates, which is ultimately of interest for device fabrication. We also note that the model further shows that the assumption of Gaussian fluctuations in the Kubo lineshape suffices to capture the experimental observation. Our measurement of electronic correlation properties provides an important experimental point of reference for quantum modeling beyond our phenomenological model. Further theoretical work may calculate these electronic fluctuations and hopefully provide a microscopic picture of specific phonons at specific wave vectors coupling to specific electronic states, but this extends beyond the scope of this work.

These 2DES experiments on perovskite nanocrystals reveal the formation of polarons in real time, by directly probing their impact on the optical response. The time dependence of the homogeneous linewidth extracted from the 2DE spectra is used as a sensitive observable to unambiguously distinguish coherent versus diffusive dynamics. The perovskite nanocrystals reveal dynamics consistent with dynamically disordered liquids, and inconsistent with ordered covalent solids. With direct observation of spectral diffusion on the femtosecond timescale, we report that the timescale of polaron formation is ~100 fs, consistent with ultrafast polar solvation dynamics in liquids. These polaron dynamics enable the rapid sampling of electronic configuration space, as revealed by their spectral diffusion dynamics in the electronic regime. This work hints at the importance in controlling the polaron properties for optoelectronic applications, in a similar spirit to previous efforts geared at controlling exciton properties in covalent nanocrystals, such as CdSe quantum dots. Future works may systematically investigate how these properties can be tuned with material parameters, such as chemical composition or quantum confinement, and hopefully provide links between polaron properties and device performance.

## Methods

**Synthesis of the CsPbI₃ nanocrystals.** The synthesis of these perovskite nanocrystals (NCs) followed previously reported hot-injection methods with some modifications[2,17,40]. In a three-neck flask, $Cs_2CO_3$ (0.60 mol) and diisooctylphosphinic acid (DOPA, 0.20 mol) were injected into octadecene (ODE, 10 mL), which was preheated at 100 °C. The solution was dried for 1 h at 120 °C under vacuum. The temperature was increased to 150 °C under an argon atmosphere, ensuring that all of the DOPA and $Cs_2CO_3$ reacted to form a Cs-phosphinate solution. The solution was then kept at 100 °C for future injection.

At the same time, $PbX_2$ (0.38 mmol) was dissolved in ODE (10 mL) in a three-neck flask, and the solution was dried for 1 h at 120 °C under vacuum. After complete solubilization, dried oleylamine (OLA, 1 mL) and dried DOPA (1 mL) were injected under an argon atmosphere. The temperature was then increased to 140–200 °C, and the hot Cs-oleate solution (0.8 mL) was subsequently injected. After 5 s, the flask was cooled down to room temperature by an ice-water bath. Depending on the temperature of injection, a strong light red (140 °C) to dark-red (200 °C) emission was observed. The product was then centrifuged for 5 min at 12,000 rpm, and the excess lead iodide was discarded. The supernatant was mixed with tert-butanol (t-BuOH) (volume ratio of 2:1 with ODE) and centrifuged for 25 min at 12,000 rpm. The supernatant was then discarded in order to remove the excess of ligands, and the NCs were re-dispersed in toluene under an ambient atmosphere and stored, in the absence of light, at 2 °C for characterization.

Fourier-transform infrared spectroscopy (FTIR) data were acquired with a Spectrum II Perkin spectrometer to confirm the lack of unbound ligands. A small amount of the NC solution was deposited onto a KBr substrate, and measured in the wavenumber range of 400–4000 $cm^{-1}$. Band attribution was made according to previous vibrational analysis[41]. An example FTIR spectrum is shown in Supplementary Fig. 1. The signal is closely related to a mixture of the toluene and ODE without additional bands. This shows that all the excess ligands have been removed during the second purification step.

Prior to the 2DES experiments, the sample was bubbled with argon for at least 30 min. This was observed to prevent degradation over the timescale of the experiments (2–4 h for a complete set). During the experiments, the sample was flown in a 0.2-mm-thick cell (Starna) with a peristaltic pump to ensure that the laser hits fresh sample.

**X-ray diffraction.** Powder X-ray diffraction data (PXRD) were measured with a Bruker D8 diffractometer by using a Cu source (Kα = 1.54˚ A) at 40 kV, 40 mA. Samples were prepared by dropping 50 μL of NCs in toluene on a glass wafer and heating the wafer to evaporate the toluene. Background data from a washed glass wafer were subtracted from raw data, and peaks were then filtered with a numerical method to suppress the noise.

**Transmission electron microscope (TEM).** TEM measurements were performed on a Jeol JEM-2100F analytical transmission electron microscope equipped with energy-selective X-ray spectrometer for chemical analysis. The samples were prepared by placing a drop of diluted CsPbI₃ nanocrystal solution on formvar-coated Cu TEM grids. An exemplary TEM image is shown in Supplementary Fig. 2.

**Time-resolved photoluminescence.** Time-resolved photoluminescence (t-PL) measurements were conducted by utilizing a streak camera (Axis TRS, Axis Photonique Inc.) in order to estimate the average number of excitations per particle as a function of fluence. The sample, dispersed in toluene, is bubbled with argon for 30 min prior to the experiment and flowed in a 0.5-mm path-length flow cell (Starna Type 48) by using a peristaltic pump (Masterflex 77390-00). The sample is excited with pulse fluences in the range of 1.1–1000 μJ cm⁻², at 3.1 eV, with a duration of <100 fs generated by frequency doubling of the output of a Ti:sapphire regenerative amplifier (Coherent Legend Elite Duo HE+, 1 kHz repetition rate) in a 100-μm BBO crystal. The fluorescence is collected at 90˚, collimated, and subsequently focused onto the streak camera slit by using a pair of off-axis parabolic mirrors. The PL is dispersed by a spectrometer (Acton SP-2358i, 150 g/mm, 600-nm blaze), accelerated in a bilamellar streak tube (Photonis P820), and imaged by using an air-cooled CCD (Spectral Instruments 1200 series). The streak tube is electronically triggered with a range of ~50 ns. Overall, the average time and energy resolution of the traces are 0.1 ns and 2.5 meV. The trace and its corresponding background measurement is obtained from an average of ten exposures lasting 10 s each.

The final trace is obtained with minimal post-processing procedures. Namely, a 1% shear correction is applied to correct for electrode and detector alignment. The value for this is obtained from traces collected with a static bias. The intensity is then corrected for uneven time bins, followed by conversion from wavelength to energy (including the Jacobian correction). Due to the simultaneous time and energy resolution of this technique, the kinetic transients extracted at each fluence come from a sum of the intensity values between 1.65 and 2.0 eV. Pump scatter at 3.1 eV is used to determine the instrument response function. Gaussian fitting of this signal results in an IRF of 200 ps. Supplementary Fig. 3 shows a representative kinetic transient with an average number of excitation $\langle N \rangle \ll 1$. The decay kinetics are well described by a triexponential, yielding an average lifetime of 17.25 ns, which is consistent with reported values for similar-sized CsPbI₃ nanocrystals[42,43]. Supplementary Note 1 provides details on how the average number of excitations per particle is estimated from these t-PL measurements. Supplementary Fig. 6 shows the results of the analysis.

**Two-dimensional electronic spectrometer.** The instrument consists of a pulse-shaper-based 2DE spectrometer operating in the pump–probe geometry and has been described in detail elsewhere[24,44,45]. Briefly, a small fraction of a Ti:Sapph amplifier (Legend Elite Duo, 8 mJ, 1 kHz, pulse duration 130 fs) is focused into a hollow-core fiber filled with argon to produce a broadband continuum. A short-pass filter is used to select the visible part of this continuum, yielding 46-μJ pulses spanning the 550–700-nm range. The visible pulses are then sent into GRISMs (GRatings + prISMs), which pre-compensate for the large dispersion arising from the pulse shapers where the 2DES pulse sequences are produced. As the pulses exit the GRISMs, they are split into a pump arm and a probe arm. The pulse shaper (Dazzler, Fastlite) in the pump arm is used to create the first two pulses in the sequence. A second pulse shaper placed in the probe arm is used to create the third and last pulse in the sequence (the probe). The experiment is carried out by varying time delay $t_1$ and taking the Fourier transform, yielding the first energy axis $E_1$. The Fourier transform of time delay $t_3$ is performed by a spectrometer, yielding the second energy axis $E_3$. Thus, one obtains a 2D correlation map $S(E_1, t_2,$ and $E_3)$, and the experiment is repeated for various population times $t_2$.

The spectral phase of the pulses is measured with a transient-grating frequency-resolved optical gating (TG-FROG) setup in a dispersion-free line[46]. The spectral phase is optimized by using the phase mask of the pulse shapers until near-Fourier-transform-limited pulses are obtained. An example TG-FROG trace acquired on the same day as the data shown in this paper is displayed in Supplementary Fig. 4.

An advantage of performing 2DES in the pump–probe geometry is that the rephasing and non-rephasing 2D signals are emitted together in the probe direction, enabling one to directly obtain an absorptive 2D spectrum exempt from phase distortions[22]. However, undesired standard pump–probe contributions are also emitted in the probe direction. By using the technique of phase cycling, as in nuclear magnetic resonance spectroscopy, it is possible to eliminate these pump–probe signals since they do not depend on the relative phase between pulses 1 and 2.

It was shown that a two-step phase cycling is sufficient to isolate the 2D signals in the pump–probe geometry[47,48]. However, in the case of highly scattering

samples, contamination to the 2D spectra can arise from pump–probe or pump–pump scatter terms. In order to minimize scatter contributions from the large CsPbI$_3$ nanocrystals, we performed a four-step phase-cycling scheme to remove pump–probe scatter contributions. The desired signal $S_{2D}$ was then given by[49]

$$S_{2D} = S(\varphi_1 = 0, \varphi_2 = 0) - S(\varphi_1 = 0, \varphi_2 = \pi) + S(\varphi_1 = \pi, \varphi_2 = \pi) \\ - S(\varphi_1 = \pi, \varphi_2 = 0) \qquad (1)$$

where $\phi_1$ and $\phi_2$ are the constant phases of pulses 1 and 2, respectively.

The CsPbI$_3$ data were acquired in a rotating frame at 0.35 PHz, which enabled the sampling of the $t_1$ axis with steps of 3 fs. This enabled the rapid acquisition of the 2D data to ensure that no sample degradation was taking place during the measurements.

The overlap of the laser spectrum with the linear absorption spectra of the samples is shown in Supplementary Fig. 5. The experimental parameters for the three samples are summarized in Supplementary Table 1. The 2DES experiments were repeated on different days with independently synthetized batches of CsPbI$_3$ nanocrystals. The lineshape behavior was reproduced every time, as exemplified in Supplementary Fig. 7.

## Data availability
The data supporting the findings of this study are available upon request.

## Code availability
The code employed for modeling the 2D spectra is open source and available at https://github.com/spalato/Mbo.jl.

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

## Acknowledgements
We gratefully acknowledge funding from the Canadian Foundation for Innovation (CFI), the Natural Sciences and Engineering Research Council of Canada (NSERC), and McGill University. Samuel Palato thanks NSERC for financial support. Hélène Seiler thanks the Swiss National Science Foundation for financial support, as well as Pierre-François Duc and Catherine Seiler for help on formatting of figures.

## Author contributions
The experiments were conducted by H.S., S.P., C.S., and H.B. Synthesis and characterization was done by E.S. and D.S. Analysis and simulations were done by H.S. and S.P. The paper was written by H.S. and P.K. with input from all authors. The research was supervised by P.K.

## Competing interests
The authors declare no competing interests.
