## [Peer Review File · Nature Communications]

Reviewers' comments:

Reviewer #1 (Remarks to the Author):

The paper by Seiler et al. reports on an experimental study, by two-dimensional electronic spectroscopy (2DES), of ultrafast non-equilibrium carrier dynamics in CsPbI₃ perovskite nanocrystals. By 2DES the authors claim to be able to detect the ultrafast structural dynamics, obtaining a direct signature of polaron formation following photoexcitation. The nanocrystals used in these studies are in the weak quantum confinement regime, so that their optical response resembles that of a bulk semiconductor, with weakly bound excitons (see absorption spectrum in Fig. 1a).

The 2DES maps correlate excitation with detection frequency. As expected, in Fig. 2b the map at early times ($t_2 = 20$ fs) is elongated along the diagonal, since electron-hole pairs excited at a given energy will Pauli block the transition at that specific energy. The map evolves over time and at $t_2 = 600$ fs it becomes elongated along the horizontal direction, as seen in Fig. 2c. In a semiconductor, the broadband excitation pulse (red spectrum in Fig. 1a) creates electron-hole pairs with significant excess energy above the bandgap. Following photoexcitation, the carriers will first thermalize with each other (carrier-carrier scattering) and then equilibrate with the cold lattice (carrier-phonon scattering or carrier cooling). Carrier cooling in lead halide perovskites occurs with a typical time constant of 200 fs at low carrier densities, which then increases for higher densities (see e.g. Price et al. Nat Commun. 6, 8420 (2015)). This time constant is consistent with the timescale of evolution of the 2DES maps observed by the authors and nicely explains, in my opinion, the evolution of the 2DES maps and the broadening of the anti-diagonal peak without invoking any polaron formation process. On the other hand the CdSe nanocrystals are in a strongly quantum confined regime and have clearly resolved excitonic transitions, therefore excitation of the bandgap exciton (as shown in Fig. S5 of the Supporting Information) would result in negligible carrier dynamics as the excitons are already at bandgap.

In this paper the authors are comparing very different systems: a quasi-bulk semiconductor, a strongly quantum confined semiconductor (with atomic-like transitions) and a molecule in solution. This comparison is in my opinion not very informative. Based on the experimental data presented, the results can be explained quite easily in terms of carrier cooling, which is however completely ignored by the authors, and I do not see in the results any direct optical signature of polaron formation. For this reason, the explanation offered by the authors does not convince me and I cannot recommend publication of this paper in Nature Communications.

Reviewer #2 (Remarks to the Author):

This manuscript reports coherent two-dimensional electronic spectroscopy on cesium lead-iodide perovskite nanocrystals. The principal conclusion is that the evolution of the optical response is inconsistent with electronic relaxation in semiconductors, but is more consistent with a solvation-like response. The authors view this as direct evidence of polaron formation dynamics. This is a highly timely work that is of general interest in the hybrid semiconductor community, and therefore has context in Nature Communications. However, my opinion is that this manuscript requires more development and perhaps analysis before publication. I strongly encourage the authors to consider the following comments.

1. (Minor comment) "Nanocrystals enables working in the single excitation per particle regime, thereby avoiding effects arising from electron-electron or hole-hole scattering processes". It is possible to observe multiple-exciton scattering signatures. See Moody et al., PHYSICAL REVIEW B 83, 115324 (2011), last figure and last paragraph.

2. In the discussion of Fig 3d and e, the authors suppose that there are no underlying crosspeaks. I don't see why the homogeneous linewidth should oscillate with time at the frequency of the LO phonon, but cross peaks do indeed. As the authors suggest, the increase in homogeneous linewidth could simply arise from spectral diffusion. I encourage the authors to elaborate substantially this discussion to clarify this argument.

3. "As expected, the width is modulated at the LO phonon frequency owing to strong exciton-coherent phonon coupling". Following from the comment above, it would strengthen the discussion to elaborate why this is an expected result. It is not clear to me.

4. It would be helpful to quantify the uncertainty in the linewidth plots.

5. "These data reveal that perovskite nanocrystals undergo spectral dynamics that are consistent with liquids and inconsistent with covalent solids." In my opinion, this is an important message of this manuscript. I would encourage the authors to rewrite this paragraph to emphasize this point more clearly. In my view, linking the solvation-like response to polaron formation dynamics makes the most sense to me.

6. It would be helpful to show the antidiagonal cuts. My guess is that fitting simple Lorentzian functions would give poor fits, so that the FWHM was considered. Given the close proximity between the antidiagonal and diagonal linewidths, I would encourage the

authors to carry out a more rigorous fit of the optical lineshape using the models in Slemens et al., *Optics Express* 18, 17699 (2010) and Bristow et al., *J. Phys. Chem. B* 115, 5365 (2011).

Reviewer #3 (Remarks to the Author):

Lead-Halide perovskites quantum dots have been examined with two-dimensional electronic spectroscopy (2DES) to probe the relationship between the optical response and the structural dynamics. The authors conclude that ultrafast dynamics on ~100fs timescale is consistent with polaron formation.

Overall, the results are well presented in this manuscript. However, the main conclusions of the paper are not strongly supported by the results. Moreover, the present state of the field is not well reflected in the bibliography. Thus, I do not recommend this paper for publication in *Nature Communications*. The main technical issues are the following:

1. In the beginning, the authors state that “Collectively, these methods reveal a picture consistent with structural dynamics following photo-excitation (6–10). The dynamics have been assigned to polaron formation and described in terms of a phonon glass with a localized ultrafast response to optical excitation (6, 7). What these structural probes do not observe is the consequence of lattice dynamics upon the optical response of the system.”

I would strongly disagree with this statement. There have been several recent high profile publications underpinning the time-resolved polaron formations with deep insights on the connection between structural ultrafast dynamics and optical response, obtained with state of the arts spectroscopies and supported by direct quantum-mechanical modeling of these systems.

(a) <https://doi.org/10.1038/s41467-018-04946-7> (resonant time-resolved Raman)

(b) <https://doi.org/10.1038/s41563-018-0262-7> (high-resolution resonant impulsive stimulated Raman spectroscopy (RISRS))

Consequently, the results of the current study are not properly placed in the context of previous works such as the above papers.

2. The main technical critique: by comparing 2D spectroscopy results for (i) perovskites, (ii) CdSe quantum dots and (iii) molecular dye, the authors conclude that the observed dynamics is qualitatively different from CdSe dot and is slow compared to the molecular system (page 4). Further analysis of the anti-diagonal linewidth indicates a diffusive dynamics in perovskites, lack of spectral diffusion in CdSe and again diffusive dynamics in the molecular system (albeit modulated by a higher frequency vibration).

I do agree with the above.

However, why in the world, the diffusive photoexcitation dynamics is necessarily called a “polaron formation”? The cases of dissipative dynamics vs oscillatory wavepacket dynamics are illustrated in the cartoons (Fig. 4 bd), which are just the cartoons, no more. Does molecular dye necessarily feature polarons (loosely defined as a charge stabilized by the lattice distortions)? I would say this would rather be a dissipative molecular exciton-vibrational dynamics rather than a charge localization by a lattice distortion.

The authors further analyze the observed dynamics in perovskites based on an empirical simplistic modeling of the multimode Brownian oscillator model. Their conclusion: “the CsPbI₃ dynamics are captured by a dissipative, diffusive model is consistent with polaron formation” is hand-waving and inconclusive. I do not see any strong case for a title “Direct optical signature of polaron formation in CsPbI₃ perovskite nanocrystals”, this is simply not the case.

3. Broad interest aspect. As authors state “Ultimately it is the optical and electronic response which is of interest for applications.” Jargon and labels apart, what new do we learn from the present study? I see a single conclusion: perovskites have the diffusive photoexcitation dynamics, which occurs on the timescale of about 100fs. This message is not particularly new and there is no detailed information on specific structural or electronic aspects. Such ultrafast relaxation timescales have been already given in tens of perovskite papers reporting time-resolved spectroscopic probes.

In comparison, already published papers such as (a) and (b) above bring a wealth of structural information such as identification of very specific vibrational motions coupled to the electronic systems and the respective fundamental changes of the electronic and optical signatures followed by a polaron formation. This is rationalized by true ab initio atomistic simulations (not a phenomenological model!) supporting experimental spectroscopy in more detail.

Consequently, I do not see how this article addresses an important broad interest aspect. What are the consequences of the observed phenomena? The sharper conclusions on how these findings

possibly facilitate physics discovery and help bringing the field toward applications, are necessary to justify placement of this work to Nature Communications.

To conclude, the present paper presents solid technical information, which deserves to be disseminated on the mainstream journals such as PCCP/JPC/Phys Rev, etc, after a proper revision. However, this work does not stand to stringent criteria of Nature Communication.

Reviewer #1 (Remarks to the Author):

Reviewer 1 has the main question of whether these results can be simply due to carrier thermalization or cooling. This question is natural as carrier relaxation processes are ubiquitous, and simpler transient absorption methods are used to unravel carrier relaxation dynamics. We did so ourselves for a decade on quantum dots, using more advanced variations of transient absorption (TA) methods. There is a key distinction, however. In these current experiments, we use **2D** spectroscopy and not 1D spectroscopy. 2D spectroscopy **uniquely** enables measurements of homogeneous and inhomogeneous linewidths. To the best of our knowledge, we are the first to apply these 2D methods to resolve the key lineshapes dynamics in lead-halide perovskites. Our experiment reveals an isomorphism between solvation dynamics of molecular dyes in polar solvent and dynamics in these ionic crystals. We assign these dissipative dynamics to polaron formation. Such observations are simply impossible using 1D spectroscopies like TA. Hence these questions cannot even be considered based upon conventional 1D TA experiments, such as our own prior works. Having said that, carrier cooling effects are indeed present. Thus Reviewer 1 has raised very valid points, which have helped us push our data analysis further. Thanks to a more in-depth analysis, we have found direct evidence that our data is **inconsistent with the signatures of cooling** previously reported in the literature, such as in the work of Price et al. Nat Commun. 6, 8420 (2015). We offer the proof in the detailed answer below as well as in the significantly revised manuscript.

COMMENT:

The paper by Seiler et al. reports on an experimental study, by two-dimensional electronic spectroscopy (2DES), of ultrafast non-equilibrium carrier dynamics in CsPbI₃ perovskite nanocrystals. By 2DES the authors claim to be able to detect the ultrafast structural dynamics, obtaining a direct signature of polaron formation following photoexcitation.

The nanocrystals used in these studies are in the weak quantum confinement regime, so that their optical response resembles that of a bulk semiconductor, with weakly bound excitons (see absorption spectrum in Fig. 1a).

The 2DES maps correlate excitation with detection frequency. As expected, in Fig. 2b the map at early times ($t_2 = 20$ fs) is elongated along the diagonal, since electron-holes pairs excited at a given energy will Pauli block the transition at that specific energy. The map evolves over time and at $t_2 = 600$ fs it becomes elongated along the horizontal direction, as seen in Fig. 2c. In a semiconductor, the broadband excitation pulse (red spectrum in Fig. 1a) creates electron-hole pairs with significant excess energy above the bandgap. Following photoexcitation, the carriers will first thermalize with each other (carrier-carrier scattering) and then equilibrate with the cold lattice (carrier-phonon scattering or carrier cooling).

Carrier cooling in lead halide perovskites occurs with a typical time constant of 200 fs at low carrier densities, which then increases for higher densities (see e.g. Price et al. Nat Commun. 6, 8420 (2015)). This time constant is consistent with the timescale of evolution of the 2DES maps

observed by the authors and nicely explains, in my opinion, the evolution of the 2DES maps and the broadening of the anti-diagonal peak without invoking any polaron formation process.

RESPONSE:

Reviewer 1 proposes that the observed lineshape signatures can be more simply explained by carrier cooling, as opposed to polaron formation. We agree that this point should have been discussed in the initial submission, and we are grateful to Reviewer 1 for pointing this out. We will now show why our data is inconsistent with carrier cooling, thanks to an extension of our previous analysis.

We first recall the basic observation of cooling in thin film perovskites, such as in the work of Price et al.. The key observation is reproduced in Figure R1 for the sake of convenience. The cooling dynamics were obtained following pumping the sample with several hundreds of meV excess energy. The high energy tail of the transient-absorption spectrum is assumed to be a hot Boltzmann distribution, from which a temperature can be extracted. In their low fluence regime, which is most similar to the regime in our work, they extract a time-constant of 230 fs. Given the temporal duration of their pump (~ 200 fs), we agree with Reviewer 1 that the timescales

(a) Normalized transient absorption (TA) spectra of $\text{CH}_3\text{NH}_3\text{PbI}_3$, pumped at 2.25 eV (pump resolution ~ 200 fs), with an initial ($t=0$) average carrier density of $N=6.4 \times 10^{17} \text{ cm}^{-3}$ over the illuminated area. Spectral broadening at early times (before 2.5 ps) indicated hot-carrier distributions. Inset: cooling rates were obtained from a global fit of the high-energy tail of each timeslice between $E=1.7$ and 1.85 eV to a Boltzmann distribution. (b) Change in photoexcited carrier temperatures against time delay for different initial ($t=0$) carrier densities, obtained from fits as described in Fig. 1a. At densities below $\sim 6 \times 10^{17} \text{ cm}^{-3}$, starting from an initial temperature of $\sim 1,000$ K down to near room temperature, the carriers cool with a principal rate constant of ~ 230 fs. At higher carrier densities, there is a slowing of the cooling rate consistent with a hot phonon bottleneck effect, which agrees with a much higher absolute carrier temperature at short-time delays (~ 300 fs).

Figure R1: Cooling dynamics in lead-halide perovskites has been successfully observed with TA spectroscopy. Here, exemplary data are reproduced from : Price et al., Hot-carrier cooling and photoinduced refractive index changes in organic–inorganic lead halide perovskites, Nature Communications 6, 8420 (2015). These data show the expected cooling signatures.

are indeed consistent with our data.

Figure R2: Extended analysis shows pseudo-TA slices extracted from our data, which can be compared with the TA slices of earlier works focusing on cooling dynamics. Our data are inconsistent with previous signatures of cooling.

However, matching timescales are by no mean a solid proof. In order to test the cooling hypothesis in a more quantitative fashion, we extract pseudo Transient-Absorption (TA) spectra from our 2D dataset. Indeed, by selecting a given value of E_1 (pump energy), one can essentially obtain a pump-probe spectrum, albeit with a better joint spectral and temporal resolution and with the additional contributions of electronic coherences, which we neglect here.

These pseudo-TA spectra can then be analyzed in terms of Ground State Bleach, Stimulated Emission and Excited State Absorption contributions. They can therefore be compared with the cooling data of Price et al. If cooling was the dominant mechanism in our data, we would expect a qualitatively similar behaviour as in previous cooling measurements.

The result of such an analysis for a value of $E_1 = 1.86 \text{ eV}$ is shown in Figure R2. As can be seen on panel (a), one observes **the rise** of a tail to the blue of the bandedge spectral feature. **This observation is therefore fully inconsistent with the observation of cooling as discussed in previous reports such as the work of Price et al.** Importantly, similar behaviour of the spectral

projections is observed for all values of E_1 , including values lower than the bandedge. These observations offer a direct proof that the observed spectral dynamics here cannot be accounted for by cooling.

We note that at later time, which are shown in panel (b), one can observe the blue tail to slightly decrease in intensity. This behaviour is qualitatively consistent with previous TA works on carrier cooling in perovskites. In our view this reflects a competition between polaron formation and cooling, as pointed out in recent work (Evans et al., JPC, 122(25) (2018)).

In addition to these observations, we stress that our laser spectrum has been tuned to the red of the bandedge, thereby minimizing the impact of cooling. Reviewer 1 states that “*the broadband excitation pulse (red spectrum in Fig. 1a) creates electron-hole pairs with significant excess energy above the bandgap.*” We disagree with this statement, based on our laser spectrum and its overlap with the linear absorption spectrum of the sample (see Figure S5).

To summarize our point, we have examined the suggestion of Reviewer 1 that the observed dynamics result from cooling. By performing additional data analysis we have shown that our data is inconsistent with previously observed signatures of cooling.

In the revised manuscript, we have now included Figure R2 as Figure 4 in the updated manuscript. A paragraph addressing the issue of cooling has been added, starting line 111 (starting with “A closer inspection”).

COMMENT:

On the other hand the CdSe nanocrystals are in a strongly quantum confined regime and have clearly resolved excitonic transitions, therefore excitation of the bandgap exciton (as shown in Fig. S5 of the Supporting Information) would result in negligible carrier dynamics as the excitons are already at bandgap.

In this paper the authors are comparing very different systems: a quasi-bulk semiconductor, a strongly quantum confined semiconductor (with atomic-like transitions) and a molecule in solution. This comparison is in my opinion not very informative.

RESPONSE:

The Referee seems unconvinced of our discussion using comparative analysis with excitons in quantum dots and molecules in solutions. We appreciate the difficulty in seeing the connection between these systems which have little in common. In our view, it is precisely because we observe an isomorphism between two systems as different as a molecule in solution and an inorganic semiconductor nanocrystal that this comparison yields insights.

Comparisons can be drawn at various levels of details. There is no doubt that the electronic structure properties of these materials have little to do with each other. Here we are not comparing microscopic details of the electronic structure. We are comparing trends and qualitative behaviour. Similarly one may find it useful to compare the thermal conductivity of a

very bad heat conductor with that of a very good heat conductor to note the qualitative points of departure. Not to discuss the microscopic effects at the origin of these differences. Here our comparison was meant exactly in this spirit. Ultimately, we agree that it is merely a choice we make.

COMMENT:

Based on the experimental data presented, the results can be explained quite easily in terms of carrier cooling, which is however completely ignored by the authors, and I do not see in the results any direct optical signature of polaron formation. For this reason, the explanation offered by the authors does not convince me and I cannot recommend publication of this paper in Nature Communications.

RESPONSE:

We have shown above that the results are inconsistent with the carrier cooling signatures reported in previous works. Again, we are grateful to Reviewer 1 to bring this important point up, as we believe it has enabled us to significantly improve the manuscript. We hope that with the revised version of the manuscript will convince the Reviewer that this work now matches the standards of Nature Communications.

Reviewer #2 (Remarks to the Author):

COMMENT:

This manuscript reports coherent two-dimensional electronic spectroscopy on cesium lead-iodide perovskite nanocrystals. The principal conclusion is that the evolution of the optical response is inconsistent with electronic relaxation in semiconductors, but is more consistent with a solvation-like response. The authors view this as direct evidence of polaron formation dynamics. This is a highly timely work that is of general interest in the hybrid semiconductor community, and therefore has context in Nature Communications. However, my opinion is that this manuscript requires more development and perhaps analysis before publication. I strongly encourage the authors to consider the following comments.

RESPONSE:

We are pleased to read that Reviewer 2 sees our work as timely and significant and hope we can address the remaining concerns in the response below.

COMMENT:

1. (Minor comment) "Nanocrystals enables working in the single excitation per particle regime, thereby avoiding effects arising from electron-electron or hole-hole scattering processes". It is possible to observe multiple-exciton scattering signatures. See Moody et al., *PHYSICAL REVIEW B* 83, 115324 (2011), last figure and last paragraph.

RESPONSE:

We fully agree with the statement of Reviewer 2, and we thank them for this precision. Semiconductor nanocrystals are indeed famously known to support multiple-exciton within a single nanocrystal. Indeed the Kambhampati group has worked intensively on this precise topic in the case of CdSe nanocrystals (see for example P. Kambhampati, *J. Phys. Chem. Lett.* 3, 1182 (2012)).

Here, we have performed estimates of average excitation per particle as a function of fluence based on streak camera measurements. The results are shown in Figure S6. We have performed the 2DE experiment in a fluence regime where the number of excitation per particle is **on average** less than one. Strictly speaking, one should always assume a non-zero contribution from carrier-carrier scattering. But here we claim this contribution to the linewidth is negligible compared to the lattice reorganization contribution.

We have modified the sentence to: "**At low fluences**, nanocrystals enables working in the single excitation per particle regime, thereby **minimizing** effects arising from electron-electron or hole-hole scattering processes".

COMMENT:

2. In the discussion of Fig 3d and e, the authors suppose that there are no underlying crosspeaks. I don't see why the homogeneous linewidth should oscillate with time at the frequency of the LO phonon, but cross peaks do indeed. As the authors suggest, the increase in homogeneous linewidth could simply arise from spectral diffusion. I encourage the authors to elaborate substantially this discussion to clarify this argument.

RESPONSE:

We apologize for the confusion. In the case of the CdSe nanocrystals (Fig 3(e)), it is well-known that the two first exciton states are strongly coupled, yielding cross-peaks on the 2D spectrum. We have extracted the linewidth of the peak using several methods. Initially, we had done so relying on fits which included crosspeaks. For an accurate estimate of the diagonal width, fitted with a Gaussian, we have observed that crosspeaks do indeed need to be taken into account. For the anti-diagonal width of the lowest exciton, of interest here, this seems not to be the case. We note that here we report the FWHM without relying of peak fitting, as we have deemed this metric to be the most honest.

Importantly, not only crosspeaks oscillate at the LO frequency as suggested by the Reviewer. The homogeneous linewidth of the diagonal peak also oscillates at the LO frequency. We agree that this is perhaps not intuitive. The width modulation is well-captured by our Huang-Rhys model of the lineshape (two-level system coupled to an undamped vibrational mode, where no cross-peaks are involved), which was described in the Supplementary and shown in Fig. 4(c). We provide additional explanations below which hopefully clarify the observation.

To understand the width modulation, it is useful to consider the Huang-Rhys model (equations 10 to 13 in the Supplementary) in two limiting cases: broad linewidths ($\sigma = \gamma = 25$ meV) and narrow linewidths ($\sigma = \gamma = 5$ meV). The results can be seen for two population times in Figure R3. The narrow linewidths allow the resolution of the vibrational substructure, which results in the appearance of satellite peaks at $\sqrt{2}E_{LO}$ corresponding to the Franck-Condon lines (specifically, the $[-1, +1]$ and $[+1, -1]$ cross peaks). The satellite peaks are suppressed at half-periods, resulting in a reduction of the peak width. We note that this model was tested in one of our recent works (Seiler et al. JCP 149(7) (2018)).

Figure R3: Model simulations at two limits. The broad linewidth limit ($\sigma = \gamma = 25$ meV, solid) and narrow linewidth limit ($\sigma = \gamma = 5$ meV, dashed). These model calculations help understand why the homogeneous width of a peak, even if this peak lies on the diagonal, is modulated at the LO frequency. Reproduced from Seiler et al. JCP 149(7) (2018).

We hope that these explanations clarify why we can expect the width to be modulated at the LO frequency in the case of the CdSe nanocrystals. We have provided a reference to our previous work in the supplementary materials.

As for the CsPbI₃ nanocrystals, we believe much less is currently understood about the details of the electronic structure in these materials. Based on the linear spectrum and the shape of the 2D spectrum, we have decided to consider a single peak with no additional crosspeaks, since we had no obvious reason to consider distinct electronic transitions.

COMMENT:

3. "As expected, the width is modulated at the LO phonon frequency owing to strong exciton-coherent phonon coupling". Following from the comment above, it would strengthen the discussion to elaborate why this is an expected result. It is not clear to me.

RESPONSE:

We thank the Reviewer for pointing this lack of clarity. We realize our initial statement was indeed not "to be expected" at this stage of the paper. The aim of Fig. 3(e) is to show that the width is modulated at the LO frequency, which is a strikingly different behaviour from the CsPbI₃ lineshape. Later in the paper we demonstrate how the width modulation at the LO phonon frequency can be perfectly captured by a Huang-Rhys model of the lineshape, as is shown in Fig. 4(c) and discussed above. We have modified the sentence line 131, which now reads: "Instead, the width is modulated at the LO phonon."

COMMENT:

4. It would be helpful to quantify the uncertainty in the linewidth plots.

RESPONSE:

We are not sure to understand the Reviewer's comment. Since we extract the FWHM, we do not get uncertainty on our linewidth estimate. If we were fitting the lineshape, we could indeed report the error on the fits. But this is not the case here.

COMMENT:

5. *"These data reveal that perovskite nanocrystals undergo spectral dynamics that are consistent with liquids and inconsistent with covalent solids." In my opinion, this is an important message of this manuscript. I would encourage the authors to rewrite this paragraph to emphasize this point more clearly. In my view, linking the solvation-like response to polaron formation dynamics makes the most sense to me.*

RESPONSE:

This suggestion is excellent. We have significantly modified the manuscript (abstract, added paragraph starting line 142 ("The Kubo lineshape..."), added paragraph starting line 157 ("The lineshape dynamics...")) to hopefully convey this essential point in a clearer fashion.

COMMENT:

6. *It would be helpful to show the antidiagonal cuts. My guess is that fitting simple Lorentzian functions would give poor fits, so that the FWHM was considered. Given the close proximity between the antidiagonal and diagonal linewidths, I would encourage the authors to carry out a more rigorous fit of the optical lineshape using the models in Siemens et al., Optics Express 18, 17699 (2010) and Bristow et al., J. Phys. Chem. B 115, 5365 (2011).*

RESPONSE:

We thank the Reviewer for the suggestion. We have added antidiagonal cuts for selected population times in the Supplementary information, as Figure S9.

As mentioned above, we have attempted to extract the linewidth using several ways. We have found that the qualitative behaviour of the lineshape was robust for all the ways which we have tested, for all 3 systems. We agree that the lineshape expressions provided in the works of Siemens et al., Optics Express 18, 17699 (2010) and Bristow et al., J. Phys. Chem. B 115, 5365 (2011) are more rigorous to extract quantitative values. Here, we focus on the qualitative observation. In our view, the FWHM remains the most honest and convincing quantity to report on the trend because it does not rely on any fit.

Reviewer #3 (Remarks to the Author):

Reviewer 3 had several questions about our initial submission. First, there was the issue of previous works which have already established how the optical response couples to specific lattice modes. Second, the Reviewer also noted that we do simple modeling and not ab initio quantum dynamics calculations. Third, the Reviewer was also unconvinced of the broader significance of our work. The first two statements are true but are not relevant to the merit or impact of our work. The third statement is false, and we will now show why. Polarons are electron-lattice phenomena. One aims to understand both the lattice and optical properties with complementary methods. The polarons in these perovskites are of interest not for their own sake but for their role in optical and electronic properties of these materials. Hence the electronic response is just as important as the lattice response. With that, these electronic measurements are the first of their kind. They can simply not be measured using 1D methods such as TA. To the best of our knowledge, no other 2DE measurement has reported on lineshape dynamics in lead halide perovskites. These dynamics are important as they uniquely and directly reveal an isomorphism between solvation dynamics and the diffuse dynamics in these ionic semiconductor nanocrystals. While previous measurements may have postulated this isomorphism, ours directly show it. We address the specific concerns of Reviewer 3 in the detailed response below.

COMMENT:

Lead-Halide perovskites quantum dots have been examined with two-dimensional electronic spectroscopy (2DES) to probe the relationship between the optical response and the structural dynamics. The authors conclude that ultrafast dynamics on ~100fs timescale is consistent with polaron formation.

Overall, the results are well presented in this manuscript. However, the main conclusions of the paper are not strongly supported by the results.

RESPONSE:

We thank Reviewer 3 for their helpful comments, which have enabled us to improve the manuscript significantly. We have realized that some of our conclusions in the initial manuscript were not sufficiently discussed. We have significantly revised the manuscript, and hopefully clarified the steps leading to our conclusions.

COMMENT:

Moreover, the present state of the field is not well reflected in the bibliography.

RESPONSE:

We fully agree with Reviewer 3 on this point and apologize for not including key references in the initial submission. Since Nature Communications does not restrict the number of papers to

cite, we have now included more references (References 11, 12, 30, 34-39 in the revised manuscript). While it is difficult to keep track of the perovskite literature for obvious reasons, we fully acknowledge that this was an important omission from our side.

COMMENT:

Thus, I do not recommend this paper for publication in Nature Communications. The main technical issues are the following:

1. In the beginning, the authors state that “Collectively, these methods reveal a picture consistent with structural dynamics following photo-excitation (6–10). The dynamics have been assigned to polaron formation and described in terms of a phonon glass with a localized ultrafast response to optical excitation (6, 7). What these structural probes do not observe is the consequence of lattice dynamics upon the optical response of the system.”

I would strongly disagree with this statement. There have been several recent high profile publications underpinning the time-resolved polaron formations with deep insights on the connection between structural ultrafast dynamics and optical response, obtained with state of the arts spectroscopies and supported by direct quantum-mechanical modeling of these systems.

(a) <https://doi.org/10.1038/s41467-018-04946-7> (resonant time-resolved Raman)

(b) <https://doi.org/10.1038/s41563-018-0262-7> (high-resolution resonant impulsive stimulated Raman spectroscopy (RISRS))

Consequently, the results of the current study are not properly placed in the context of previous works such as the above papers.

RESPONSE:

Reviewer 3 raises the valid point that previous works have investigated the relationships between optical response and specific phonon modes, and have linked their observations to polarons. These previous studies are indeed relevant and should have been cited. In the updated manuscript, they are included as References 11 and 12. However, the novelty of our work is not affected by these previous works, as they measure different things.

The connection between the vibrational and electronic structures in lead-halide perovskites systems is complex. The dream experiment would be one where the couplings of phonons to the electronic system across the whole Brillouin zone are revealed with mode specificity. While methods such as Time-resolved Angle-Resolved Photo-Emission or Femtosecond Electron Diffraction go towards this direction, the current state of research is simply not there yet. Put simply: neither time-resolved Raman or two-dimensional spectroscopies are therefore able to provide the full picture.

Having said that, we believe that the works cited by Reviewer 3 and our work each bring important pieces of the puzzle, and are complementary to each other. Indeed, while the works

by the groups of Carlo Silva and Richard Mathies are indeed phonon mode specific, they are restricted to zone-center phonons by definition (assuming higher order Raman signals are neglected). The offered picture is thus far from comprehensive, since phonons are defined across the whole Brillouin zone. We note that this statement is valid whether a polaron is invoked or not. But it is especially pertinent in the case of polaron formation, since it “results from charge coupling to higher-frequency longitudinal optical (LO) phonon modes at higher momentum.” (X.Y. Zhu, Nature Materials 17, 379–381 (2018)). These higher momenta phonons cannot be accessed by time-resolved Raman methods.

In contrast to Raman spectroscopy, our work is not phonon specific. Our method is sensitive to the net impact of structural response on the optical response, which is complementary and equally important. In that sense we do not think that the novelty of our work is in any fashion affected by previous time-resolved Raman works. Both works employ advanced methods to look at the problem differently.

Reviewer 3 also mentions that previous time-resolved Raman works were “supported by direct quantum-mechanical modeling of these systems”. The observations were indeed rationalized by DFT, as opposed to a phenomenological model in our case. We note, however, that it is not because quantum-mechanical modeling is employed to accurately describe specific atomic motions that this modeling accurately describes the process of polaron formation. The theoretical modelling of coupled electronic and structural dynamics in perovskites systems, which feature very soft phonon modes and large anharmonicities, remains a topic of current investigation due to its complexity. While certainly informative, it is therefore highly unlikely that the quantum-mechanical modeling within the harmonic approximation and restricted to a very narrow part of the Brillouin zone ($k = 0$), can fully capture the polaron formation process. While we of course agree that a microscopic modelling of our observations would be desirable in principle, we actually see value in phenomenological modelling given that a comprehensive microscopic theory of polaron formation is yet to be established. We have now made this point, line 168 of the revised manuscript (sentence starting with “Our measurement of electronic correlations properties...”).

To summarize, we agree that previous works have investigated the relationships between optical response and specific phonon modes. These works have provided important insights, but we are convinced that they do not affect the novelty of our work in any fashion as these experiments look at different aspects of a complex problem.

COMMENT:

2. The main technical critique: by comparing 2D spectroscopy results for (i) perovskites, (ii) CdSe quantum dots and (iii) molecular dye, the authors conclude that the observed dynamics is qualitatively different from CdSe dot and is slow compared to the molecular system (page 4). Further analysis of the anti-diagonal linewidth indicates a diffusive dynamics in perovskites, lack

of spectral diffusion in CdSe and again diffusive dynamics in the molecular system (albeit modulated by a higher frequency vibration).

I do agree with the above.

However, why in the world, the diffusive photoexcitation dynamics is necessarily called a “polaron formation”? The cases of dissipative dynamics vs oscillatory wavepacket dynamics are illustrated in the cartoons (Fig. 4 bd), which are just the cartoons, no more. Does molecular dye necessarily feature polarons (loosely defined as a charge stabilized by the lattice distortions)? I would say this would rather be a dissipative molecular exciton-vibrational dynamics rather than a charge localization by a lattice distortion.

RESPONSE:

Reviewer 3 is not convinced about the terminology employed, namely that our observation should be called polaron formation. We address this concern below. However, terminology should not distract from the main point. The main point is that this work is the first to directly reveal that the electronic homogeneous lineshape in an ionic semiconductor nanocrystal resembles solvation in a polar liquid.

Should our observation be called “a dissipative molecular exciton-vibrational dynamics” instead of a polaron as suggested by the Reviewer? We are not convinced that this is a better choice of words, since the observed dynamics takes place in an inorganic bulk-like semiconductor nanocrystal, as opposed to a molecular system. Hence the diffuse dynamics must involve the lattice degrees of freedom in some fashion. That being said, we agree that the term polaron is loosely defined. In fact, when analyzing the data, we were initially reluctant to employ this term as we thought it may be “superfluous jargon”. We have finally decided to adopt this terminology for the following reasons:

- 1) **Polaron formation is conceptually analogous to solvation:** we note that this claim does not come from us. It has been made, for example, by the Photoemission community following decades of extensive studies of polaron dynamics in molecules on surfaces (See for example Bovensiepen et al.: Dynamics Solid State Surfaces and Interfaces: Volume 2: Fundamentals (Wiley-VCH, 2012)). More topical, the group of XiaoYang Zhu, worldleading expert on perovskites, has also recently made the connection based on similar signals observed in optical Kerr effect experiments (Zhu et al., Science. 353, 1409–1413 (2016)). This conceptual connection is indeed natural to us, as both phenomena arise from atomic fluctuations coupling to electronic degrees of freedom. It is far from obvious that the microscopic nature of the fluctuations are related in both cases. But in both cases it is the fluctuations which stabilize the excited carrier, yielding either a solvated electron or a polaron.
- 2) **Context of the work:** several works, including those referred to by Reviewer 3, have provided evidence for polarons in lead-halide perovskites. Indeed we do not invoke polaron out of the blue. We note that if the usage of the term polaron is questioned in

the context of our work, we do not understand why it should not be questioned in other works. Given the coupled nature of this quasiparticle, it is always observed either from the atomic or electronic side and never directly (we have changed the title of our manuscript for this reason).

These two points justify, in our view, the assignment of the observed dynamics to a polaron. We believe this is the most suitable term given the key observation and the current state of knowledge.

COMMENT:

The authors further analyze the observed dynamics in perovskites based on an empirical simplistic modeling of the multimode Brownian oscillator model. Their conclusion: “the CsPbI3 dynamics are captured by a dissipative, diffusive model is consistent with polaron formation” is hand-waving and inconclusive. I do not see any strong case for a title “Direct optical signature of polaron formation in in CsPbI3 perovskite nanocrystals”, this is simply not the case.

RESPONSE:

The Reviewer does not think that we have convincingly justified the claim of polaron formation in our initial submission. This comment shows that a more in-depth discussion is needed. We have done so in the revised version of the manuscript (in particular, see paragraph added starting line 142, “The Kubo lineshape”) and below. **We explain why 2DE is by definition well-suited to investigate the problem of polaron formation.**

Polaron formation arises from the dynamical coupling of atomic structure fluctuations to electronic states. It is these fluctuations which yield charge localization. The influence of the dynamical coupling on the optical response translates in a time-dependence of the electronic gap. These are all well-known general facts of polaron theory and are described in numerous textbooks and reviews (For example Stoneham et al. J. Phys. Condens. Matter. 19, 255208 (2007)).

These fluctuations are difficult to measure. Indeed fluctuations of the electronic gap on the femtosecond timescale cannot be resolved directly in the time domain for obvious reasons. But their correlations can be measured in the frequency domain via the frequency-frequency correlation function (FFCF). This is precisely what is measured in a 2D experiment, and we refer the Reviewer to reference textbooks for more details (For example Hamm and Zanni , CUP (2011)). We note that one of the key historical motivations to develop 2D spectroscopies was to have access to these FFCF. Because polaron formation arises from fluctuations and that 2DE is an ideal method to investigate the correlations of these fluctuations, we can make the general statement that the 2DE method is well-suited to investigate polaron dynamics in solids via the FFCF.

As a case in point, polaron signatures in 2DE spectroscopy have been theoretically and experimentally investigated for different systems (Huynh et al., JCP, 139(10) (2013), Dahlbom et

al., CPL, 364(6) (2002)). Our observations are fully consistent with these signatures. This fact was not stated clearly in the initial submission. We hope that the added paragraphs now clarify the key ideas.

The Reviewer also takes issue with our simple modelling. Again, we stress that the experimental observable of our work is the highlight, not the modelling. Our experiment is able to provide information about the electronic fluctuations (their correlation time and their amplitude), which are non-trivial to compute from the quantum modelling side.

We hope that the revised title as well as the clarifying paragraphs have convinced Reviewer 3 that the method of 2DES is well-suited to investigate polarons, just like it is well-suited to observe solvation.

COMMENT:

3. Broad interest aspect. As authors state “Ultimately it is the optical and electronic response which is of interest for applications.” Jargon and labels apart, what new do we learn from the present study? I see a single conclusion: perovskites have the diffusive photoexcitation dynamics, which occurs on the timescale of about 100fs. This message is not particularly new and there is no detailed information on specific structural or electronic aspects. Such ultrafast relaxation timescales have been already given in tens of perovskite papers reporting time-resolved spectroscopic probes.

RESPONSE:

The Reviewer claims that our study does not offer significant new insights in comparison with existing works. We believe this assessment is inaccurate. The statement “*Such ultrafast relaxation timescales have been already given in tens of perovskite papers reporting time-resolved spectroscopic probes*” is misleading. It is not because other works have measured similar timescales that they have measured the same quantity. **One-dimensional spectroscopies cannot measure homogeneous lineshape dynamics.** Our measurement is, to our knowledge, the first to reveal in an unambiguous fashion diffuse dynamics in the electronic response of lead-halide perovskites.

We believe this conclusion has important implications, which have little to do with jargon and labels. The field of perovskites has emerged rapidly over the recent years. However it is plagued by an interplay of very complex processes and a lack of experiments that can delineate these complex processes. Here, we apply a state-of-the-art spectroscopy to probe these materials. 2DE is uniquely designed to observe homogeneous lineshape dynamics. From these lineshape dynamics, we extract quantities which cannot be extracted with one-dimensional electronic spectroscopies, such as TA. In reworking the manuscript we have also realized that aside from the polaron formation timescale, another important quantity can be directly extracted from the amplitude of the fluctuations: the reorganization energy. This quantity is important because it is

a measure of polaron stability. This quantity plays an important role in the carrier dynamics of lead-halide perovskites, which has obvious implications for opto-electronic devices.

COMMENT:

In comparison, already published papers such as (a) and (b) above bring a wealth of structural information such as identification of very specific vibrational motions coupled to the electronic systems and the respective fundamental changes of the electronic and optical signatures followed by a polaron formation. This is rationalized by true ab initio atomistic simulations (not a phenomenological model!) supporting experimental spectroscopy in more detail.

RESPONSE:

We believe we have addressed this point above.

COMMENT:

Consequently, I do not see how this article addresses an important broad interest aspect. What are the consequences of the observed phenomena? The sharper conclusions on how these findings possibly facilitate physics discovery and help bringing the field toward applications, are necessary to justify placement of this work to Nature Communications.

To conclude, the present paper presents solid technical information, which deserves to be disseminated on the mainstream journals such as PCCP/JPC/Phys Rev, etc, after a proper revision. However, this work does not stand to stringent criteria of Nature Communication.

RESPONSE:

We hope that our revised manuscript has addressed the concerns of Reviewer 3 and convincingly demonstrates the significance of our observation. Regardless of whether we call it a polaron or not, we have revealed for the first time a surprising isomorphism between the electronic lineshape dynamics of an ionic semiconductor nanocrystal and a molecular dye undergoing solvation.

REVIEWERS' COMMENTS:

Reviewer #1 (Remarks to the Author):

In their revised version of the manuscript, the authors have seriously taken into account my criticism concerning the origin of the fast dynamics observed in the 2DES maps and the possibility that it is simply due to carrier cooling, rather than the polaron formation invoked by the authors. By analyzing cross sections of the 2DES maps for a fixed excitation photon energy and different values of the waiting time t_2 , they clearly observed at early times a rise of the blue tail of the spectrum, which is inconsistent with a carrier cooling description. This satisfactorily addresses my main objection. The current version of the paper is thus in my opinion suitable for publication in Nature Communications.

Reviewer #2 (Remarks to the Author):

The authors have largely addressed my comments. I would simply add to my previous comments and to the response to comment 6 in page 9 of the response document that the antidiagonal lineshape is only a reliable representation of the homogeneous line in the rephasing measurement in the limit of strong inhomogeneous broadening. When the homogeneous and inhomogeneous contributions are comparable, as it appears to be the case for the perovskite nanocrystals, a more involved analysis of both the diagonal and antidiagonal lineshape is called for. I appreciate that the authors prefer to keep the discussion of the lineshape evolution at a comparative level, and I think that this is fine, but I would recommend that this point be addressed in the text of the manuscript.

Reviewer #3 (Remarks to the Author):

The authors made an honest attempt to answer to all three referee's comments (cooling, jargon, bibliography, etc.) and have thoroughly revised the MS including change in the title. Subsequently, the technical quality and technical content of the article have been substantially improved.

Now the question is whether this article meets the standards of Nature Communications. In my opinion, novelty and broad interest/impact criteria are still insufficiently addressed. Let take as an example the response to my report (3rd Referee) and the abstract/intro/conclusion of the article. I am here approaching from the position of a general reader of this manuscript:

“With that, these electronic measurements are the first of their kind. They can simply not be measured using 1D methods such as TA. To the best of our knowledge, no other 2DE measurement has reported on lineshape dynamics in lead halide perovskites. These dynamics are important as they uniquely and directly reveal an isomorphism between solvation dynamics and the diffuse dynamics in these ionic semiconductor nanocrystals. While previous measurements may have postulated this isomorphism, ours directly show it.”

Here we go... So what? What do we learn from this isomorphism between solvation dynamics and

the diffuse dynamics in perovskites? What is the message for a broader community?

“In contrast to Raman spectroscopy, our work is not phonon specific. Our method is sensitive to the net impact of structural response on the optical response, which is complementary and equally important.”

Why equally important? Elaborate!

“Our measurement is, to our knowledge, the first to reveal in an unambiguous fashion diffuse dynamics in the electronic response of lead-halide perovskites.”

So what?

“We believe this conclusion has important implications, which have little to do with jargon and labels. The field of perovskites has emerged rapidly over the recent years. However, it is plagued by an interplay of very complex processes and a lack of experiments that can delineate these complex processes.”

Name these implications!

“... we extract quantities which cannot be extracted with one-dimensional electronic spectroscopies, such as TA”

So what?

Let us now exemplify these issues on the current abstract/introduction/conclusion of this paper:

Abstract:

“The polaron formation time and binding energy are extracted from the data.”

“Formation time” – was reported elsewhere. Here the report confirms consistency.

“Binding energy” – the situation is similar; it is just the consistency with previous works. Indeed, a sentence from the paper: “This value is a measure of polaron stability, which is expected to play an important role in the competition between polaron formation and cooling (38). Our extracted value is on the same order of magnitude as previous cryo magneto optical absorption spectroscopy measurements and also falls within the bounds of calculated binding energies for hybrid organic-inorganic perovskites (7, 39).

Do these measurements sufficiently support novelty of this study?

Next let take a look to the claim of importance of isomorphism between solvation dynamics and the diffuse dynamics in perovskites

Abstract “these dynamics are consistent with liquid-like structural dynamics on the 100 femtosecond timescale”

Intro: “Our data show that the homogeneous linewidth of the CsPbI₃ nanocrystals evolves on the

100 fs timescale in a manner consistent with ultrafast polar solvation dynamics, contrasting with the response of covalent CdSe nanocrystal quantum dots.”

Conclusion: “The perovskite nanocrystals reveal dynamics consistent with dynamically disordered liquids, and inconsistent with ordered covalent solids. With direct observation of spectral diffusion on the femtosecond timescale, we report the timescale of polaron formation is ~ 100 fs, consistent with ultrafast polar solvation dynamics in liquids.”

Ok, this technical conclusion has been repeated multiple times. I hope now to understand its importance and implications for the broader field and emerging perovskite technologies.

So that we have the following last sentences of the article:

“These polaron dynamics enable the rapid sampling of electronic configuration space, as revealed by their spectral diffusion dynamics in the electronic regime. An understanding of the relationships between polaron induced spectral diffusion and material design may enable the rational design of new functionality.”

To me these are very generic words. What is so special and important about “rapid sampling of electronic configuration space”? Does this mean that in classic semiconductors this sampling is slow? What are “the relationships between polaron induced spectral diffusion and material design”? To me the conclusion is ‘polaron formation dynamics is carefully explored in CsPbI₃ material’ – there are no obvious relationships and relevance to materials design. Even more puzzling is, how do these observations “enable the rational design of new functionality”?

To conclude, the revised version of the MS lack answers to these questions that would help a general reader to understand an importance of the present work. As such, I cannot recommend his article for publication in Nature Comm without addressing these issues.

Reviewer #1 (Remarks to the Author):

COMMENT:

In their revised version of the manuscript, the authors have seriously taken into account my criticism concerning the origin of the fast dynamics observed in the 2DES maps and the possibility that it is simply due to carrier cooling, rather than the polaron formation invoked by the authors. By analyzing cross sections of the 2DES maps for a fixed excitation photon energy and different values of the waiting time t_2 , they clearly observed at early times a rise of the blue tail of the spectrum, which is inconsistent with a carrier cooling description. This satisfactorily addresses my main objection. The current version of the paper is thus in my opinion suitable for publication in Nature Communications.

RESPONSE:

We are glad to have addressed the initial concerns of Reviewer 1 in our revised manuscript. We thank the Reviewer again for helping us push our analysis further and strengthen the paper.

Reviewer #2 (Remarks to the Author):

COMMENT:

The authors have largely addressed my comments. I would simply add to my previous comments and to the response to comment 6 in page 9 of the response document that the antidiagonal lineshape is only a reliable representation of the homogeneous line in the rephasing measurement in the limit of strong inhomogeneous broadening. When the homogeneous and inhomogeneous contributions are comparable, as it appears to be the case for the perovskite nanocrystals, a more involved analysis of both the diagonal and antidiagonal lineshape is called for. I appreciate that the authors prefer to keep the discussion of the lineshape evolution at a comparative level, and I think that this is fine, but I would recommend that this point be addressed in the text of the manuscript.

RESPONSE:

We fully agree with Reviewer 2 that the anti-diagonal lineshape is only representative of the homogeneous linewidth in the strong inhomogeneous broadening limit. We note that in the case of nanocrystals, we can expect a large static contribution to the diagonal linewidth due to significant size dispersion (as seen in Supplementary Figure 2). However, we agree that additional details should have been added in the main text.

In the updated manuscript, we have added the following sentences, starting line 132:

“We note that the anti-diagonal linewidth is only a true representation of the homogeneous linewidth in the limit of strong inhomogeneous broadening. Due to the size dispersion of the nanocrystals (see Supplementary Figure 2), we can expect a large static disorder contribution to the diagonal linewidth of the 2D peak. Here we report the full-width at half-maximum (FWHM) of the anti-diagonal lineshape as a function of t_2 , which has the advantage to be model free but is an approximation to the true homogeneous linewidth.”

Reviewer #3 (Remarks to the Author):

Reviewer 3 seems satisfied with the technical improvements to the revised manuscript, but remains concerned about the overall significance of the results for the broader scientific community. More specifically, Reviewer 3 raises several times the question of what is learnt from the isomorphism between solvation dynamics and the diffuse dynamics in perovskites. Since this is a recurrent point, we address it here. Remaining issues are addressed below where appropriate.

We wish to stress that what is learnt in our manuscript is the isomorphism itself. This isomorphism is significant in itself, as it is far from obvious that the ultrafast electronic response of a semiconductor nanocrystals displays liquid-like properties. Such dynamics are simply not observed in covalent nanocrystals, where spectral diffusion is observed on the nanosecond timescale – a million times slower! Having said that, one can indeed ask the next question: now that this isomorphism has been revealed, what are its implications and why should a device scientist care?

A device scientist should care because different strategies have to be applied to optimize device performance based on the fundamental photo-physical processes at play in materials. As an example, the field of dye-sensitized solar cells went from employing organic dyes, to CdSe quantum dots to perovskites nanocrystals, all as sensitizers. While each of these materials are employed for the same function in the device, optimization strategies depend on the microscopic relaxation channels, which differ drastically as is precisely shown in this work and that of others.

Whereas excitonic effects are dominating in covalent semiconductors such as CdSe, our work shows that polaronic effects seem to play an essential role in electron dynamics in perovskites. Similar to efforts towards controlling excitonic properties in covalent quantum dots, the implication of our finding is therefore that the relevant properties to control in the case of perovskites are polaronic. Thus our study hints that parameters to optimize may be, for example, polaron binding energy. To what extent can it be tuned by chemical composition? How are the polaronic properties affected by quantum confinement? We believe our work enables to formulate these questions, of direct relevance to material scientists, but which go far beyond the scope of this work.

To make this message clearer, we have now removed the following sentence of the manuscript: “An understanding of the relationships between polaron induced spectral diffusion and material design may enable the rational design of new functionality.”

We have added a new paragraph, which we hope is more specific: “This work hints at the importance of controlling polaron properties for opto-electronic applications, in a similar spirit to previous efforts geared at controlling exciton properties in covalent nanocrystals, such as CdSe quantum dots. Future works may systematically investigate how these properties can be tuned with materials parameters,

such as chemical composition or quantum confinement, and hopefully provide links between polaron properties and device performance.”

COMMENT:

The authors made an honest attempt to answer to all three referee’s comments (cooling, jargon, bibliography, etc.) and have thoroughly revised the MS including change in the title. Subsequently, the technical quality and technical content of the article have been substantially improved.

RESPONSE:

We thank the Reviewer for acknowledging the improvements in the revised version of the manuscript.

COMMENT:

Now the question is whether this article meets the standards of Nature Communications. In my opinion, novelty and broad interest/impact criteria are still insufficiently addressed. Let take as an example the response to my report (3rd Referee) and the abstract/intro/conclusion of the article. I am here approaching from the position of a general reader of this manuscript:

“With that, these electronic measurements are the first of their kind. They can simply not be measured using 1D methods such as TA. To the best of our knowledge, no other 2DE measurement has reported on lineshape dynamics in lead halide perovskites. These dynamics are important as they uniquely and directly reveal an isomorphism between solvation dynamics and the diffuse dynamics in these ionic semiconductor nanocrystals. While previous measurements may have postulated this isomorphism, ours directly show it.”

Here we go... So what? What do we learn from this isomorphism between solvation dynamics and the diffuse dynamics in perovskites? What is the message for a broader community?

RESPONSE: asd

We believe we have addressed this comment at the beginning of our response to Reviewer 3’s comments.

COMMENT:

“In contrast to Raman spectroscopy, our work is not phonon specific. Our method is sensitive to the net impact of structural response on the optical response, which is complementary and equally important.”

Why equally important? Elaborate!

RESPONSE:

Phonon-specific information is key to increase our fundamental understanding of polaronics in these materials. However, in a device, the excited electron or hole couple to a wealth of different phonons simultaneously, not only zone-center phonons probed in standard Raman experiments. In fact, this is the very process by which a polaron can be formed in the first place. In order to localize the charge in real space, a polaron must comprise of a phonon wavepacket containing a combination of different phonon modes of different wavevectors and energies. Our experiment is sensitive to the net effect of all these phonons on the electronic response. In our view, this situation is therefore relevant as it

resembles what happens upon photo-excitation in a perovskite device. We are convinced that different measurements are needed to look at different faces of a complex problem. In that sense, time-resolved Raman and 2DE spectroscopies are both necessary and complement each other.

COMMENT:

“Our measurement is, to our knowledge, the first to reveal in an unambiguous fashion diffuse dynamics in the electronic response of lead-halide perovskites.”

So what?

“We believe this conclusion has important implications, which have little to do with jargon and labels. The field of perovskites has emerged rapidly over the recent years. However, it is plagued by an interplay of very complex processes and a lack of experiments that can delineate these complex processes.”

Name these implications!

RESPONSE:

We believe we have addressed this comment at the beginning of our response to Reviewer 3’s comments.

COMMENT:

“... we extract quantities which cannot be extracted with one-dimensional electronic spectroscopies, such as TA”

So what?

RESPONSE:

Polaron formation time and binding energy do matter. Just like the opto-electronic properties of excitonic materials depend significantly on exciton properties (binding energy, lifetime, ...), we can expect the opto-electronic properties of polaronic materials to depend significantly upon polaron properties. The fact that our measurement can cleanly extract quantities such as polaron formation time and binding energy therefore significantly adds value with respect to a standard TA measurement.

COMMENT:

*Let us now exemplify these issues on the current abstract/introduction/conclusion of this paper:
Abstract:*

“The polaron formation time and binding energy are extracted from the data.”

“Formation time” – was reported elsewhere. Here the report confirms consistency.

“Binding energy” – the situation is similar; it is just the consistency with previous works. Indeed, a sentence from the paper: “This value is a measure of polaron stability, which is expected to play an important role in the competition between polaron formation and cooling (38). Our extracted value is on the same order of magnitude as previous cryo magneto optical absorption spectroscopy measurements and also falls within the bounds of calculated binding energies for hybrid organic-inorganic perovskites (7,

39).

Do these measurements sufficiently support novelty of this study?

RESPONSE:

The Reviewer is concerned with the novelty of our study pertaining specifically to the extraction of the polaron binding energy, as some cyro magneto optical absorption spectroscopy measurements have previously reported a polaron binding energy. Here we draw again on the past experience on excitonic materials. Scientists have spent decades measuring exciton binding energies in semiconductors with various more or less sophisticated methods, ranging all the way from linear absorption to time-resolved ARPES measurements. The reason why so much effort is dedicated to extracting these quantities is because they matter for applications yet are typically difficult to extract. For CdSe, a case which we have studied extensively, we have learnt that the binding energy of the biexciton extracted using different methods systematically yields different values. The reason for these different values are that different techniques are sensitive to slightly different quantities. Thus the binding energy of the biexciton is found to be different when measured via time-resolved photoluminescence, transient absorption spectroscopy or 2DE spectroscopy.

In light of this, we do not see how previous studies reporting polaron binding energies using totally different methods negatively impact the novelty of this work.

COMMENT:

Next let take a look to the claim of importance of isomorphism between solvation dynamics and the diffuse dynamics in perovskites

Abstract “these dynamics are consistent with liquid-like structural dynamics on the 100 femtosecond timescale”

Intro: “Our data show that the homogeneous linewidth of the CsPbI₃ nanocrystals evolves on the 100 fs timescale in a manner consistent with ultrafast polar solvation dynamics, contrasting with the response of covalent CdSe nanocrystal quantum dots.”

Conclusion: “The perovskite nanocrystals reveal dynamics consistent with dynamically disordered liquids, and inconsistent with ordered covalent solids. With direct observation of spectral diffusion on the femtosecond timescale, we report the timescale of polaron formation is ~ 100 fs, consistent with ultrafast polar solvation dynamics in liquids.”

Ok, this technical conclusion has been repeated multiple times. I hope now to understand its importance and implications for the broader field and emerging perovskite technologies.

RESPONSE:

We believe we have addressed this comment at the beginning of our response to Reviewer 3's comments.

COMMENT:

So that we have the following last sentences of the article: “These polaron dynamics enable the rapid sampling of electronic configuration space, as revealed by their spectral diffusion dynamics in the electronic regime. An understanding of the relationships between polaron induced spectral diffusion and material design may enable the rational design of new functionality.”

To me these are very generic words. What is so special and important about “rapid sampling of electronic configuration space”? Does this mean that in classic semiconductors this sampling is slow? What are “the relationships between polaron induced spectral diffusion and material design”? To me the conclusion is ‘polaron formation dynamics is carefully explored in CsPbI3 material’ – there are no obvious relationships and relevance to materials design. Even more puzzling is, how do these observations “enable the rational design of new functionality”?

RESPONSE:

The rapid sampling of electronic configuration space (spectral diffusion on 100 fs timescale) is surprising for a semiconductor nanocrystal. This fast sampling, reminiscent of solvation in a polar solvent, would simply not happen in a covalent semiconductor. We exemplify this point with CdSe, where no spectral diffusion is observed. We emphasize again that in covalent CdSe, spectral diffusion is observed on the nanosecond timescale - one million times slower!

COMMENT:

To conclude, the revised version of the MS lack answers to these questions that would help a general reader to understand an importance of the present work. As such, I cannot recommend his article for publication in Nature Comm without addressing these issues.

RESPONSE:

We hope this response letter as well as the revised manuscript shows why this work is significant for a broader audience. Whenever a new class of materials emerges as potential candidates for optoelectronic devices, the natural question arises: what physics is new and to what extent can we build on previous knowledge acquired on other semiconductor nanocrystals? This question is highly relevant to the field of perovskites. Many scientists currently working on perovskites have worked for decades on other semiconductor nanocrystals, such as CdSe quantum dots. This applies not only to research groups investigating fundamental aspects of semiconductor nanocrystals, but also to synthesis and device groups. Thus this question is of particular relevance for the emerging perovskites community, and we believe our work contributes to provide some answers to this broadly relevant question.